# Machine Learning-Enabled Non-Invasive Screening of Tumor-Associated Circulating Transcripts for Early Detection of Colorectal Cancer

**DOI:** 10.3390/ijms26041477

**Published:** 2025-02-10

**Authors:** Jin Han, Sunyoung Park, Li Ah Kim, Sung Hee Chung, Tae Il Kim, Jae Myun Lee, Jong Koo Kim, Jae Jun Park, Hyeyoung Lee

**Affiliations:** 1Department of Biomedical Laboratory Science, College of Software and Digital Healthcare Convergence, Yonsei University Mirae Campus, Wonju 26493, Republic of Korea; kristenlovemom@gmail.com (J.H.); riah0416@naver.com (L.A.K.); 2School of Mechanical Engineering, Yonsei University, Seoul 03722, Republic of Korea; angelsy88@gmail.com; 3INOGENIX Inc., Chuncheon 24232, Republic of Korea; shchung@inogenix.com; 4Division of Gastroenterology, Department of Internal Medicine, Yonsei University College of Medicine, Seoul 03722, Republic of Korea; taeilkim@yuhs.ac (T.I.K.); jaejpark@yuhs.ac (J.J.P.); 5Department of Family Medicine, Wonju College of Medicine, Yonsei University, Wonju 26426, Republic of Korea; jaemyun@yuhs.ac; 6Department of Microbiology and Immunology, Institute for Immunology and Immunological Diseases, Yonsei University College of Medicine, Seoul 03722, Republic of Korea; kimjk214@yonsei.ac.kr

**Keywords:** colorectal cancer, tumor-associated circulating transcripts blood-based assay, cancer biomarkers, qPCR, machine learning, non-invasive cancer diagnosis, deep neural network

## Abstract

Colorectal cancer (CRC) is a major cause of cancer-related mortality, highlighting the need for accurate and non-invasive diagnostics. This study assessed the utility of tumor-associated circulating transcripts (TACTs) as biomarkers for CRC detection and integrated these markers into machine learning models to enhance diagnostic performance. We evaluated five models—Generalized Linear Model, Random Forest, Gradient Boosting Machine, Deep Neural Network (DNN), and AutoML—and identified the DNN model as optimal owing to its high sensitivity (85.7%) and specificity (90.9%) for CRC detection, particularly in early-stage cases. Our findings suggest that combining TACT markers with AI-based analysis provides a scalable and precise approach for CRC screening, offering significant advancements in non-invasive cancer diagnostics to improve early detection and patient outcomes.

## 1. Introduction

Colorectal cancer (CRC) is one of the leading causes of cancer-related mortality worldwide, and survival rates vary significantly based on early detection and timely intervention [1,2,3]. Traditional diagnostic methods, such as colonoscopy and fecal occult blood testing (FOBT), are effective but have limitations. Colonoscopy, although accurate, is invasive and leads to low compliance, whereas non-invasive options, such as FOBT, show limited sensitivity, particularly for early-stage CRC detection. Therefore, an innovative, non-invasive diagnostic tool that combines high accuracy with patient compliance is urgently needed to improve early detection outcomes.

With advances in cancer diagnostics, liquid biopsy has emerged as a promising alternative, enabling the real-time monitoring of cancer progression through minimally invasive methods. Tumor-associated circulating transcripts (TACTs), a novel class of RNA biomarkers, have been used for the early diagnosis of breast cancer (BC) [4]. Given the unique tumor biology and progression patterns of CRC, assessing the expression levels and diagnostic relevance of these TACT markers is crucial, specifically in CRC, to determine their broader applicability across cancer types.

Recent progress in cancer diagnostics has prompted the investigation of circulating biomarkers that can be detected using liquid biopsies [5,6]. TACTs include a panel of 10 markers that have previously demonstrated high diagnostic accuracy in detecting BC through RNA-based liquid biopsy. Given the biological similarities between CRC and BC, such as common epithelial-to-mesenchymal transition (EMT) pathways, we hypothesized that these markers could exhibit differential expression patterns in CRC and may possess diagnostic significance in CRC [7].

Machine learning is increasingly utilized in biomedical research to analyze complicated datasets and identify patterns that may not be evident through traditional statistical approaches [8,9,10,11]. We evaluated five distinct machine learning models—Generalized Linear Model (GLM), Random Forest (RF), Gradient Boosting Machine (GBM), Deep Neural Network (DNN), and automated machine learning algorithm (AutoML)—to determine which model could most accurately predict CRC using TACT marker data.

This study aimed to evaluate whether the 10 TACT markers, previously shown to be effective in BC, can provide similar diagnostic utility in CRC. To this end, we investigated the relevance of each marker in CRC by analyzing its blood expression levels among patients with CRC and healthy controls, thereby offering potential insights into cancer-type-specific diagnostic panels. In addition, this study sought not only to validate TACT markers for CRC but also to enhance the diagnostic performance of these biomarkers by integrating artificial intelligence (AI) and machine learning models. After assessing the sensitivity, specificity, and overall accuracy of each model, the DNN model was found to be the most effective. The DNN model exhibited enhanced performance, showing increased sensitivity and specificity relative to the other models, positioning it as a promising approach for advancing AI-driven diagnostic tools for CRC. Integrating TACT markers with AI analysis introduces a novel approach for CRC screening, presenting the possibility of a non-invasive, precise, and scalable diagnostic method that may enhance patient outcomes and lower healthcare expenses. Therefore, the results of this study may facilitate the development of a blood-based assay that improves CRC screening, especially for individuals hesitant to participate in invasive procedures.

## 2. Results

### 2.1. TACT Marker Expression in CRC Cell Lines

In this study, we first evaluated the expression of 10 TACTs previously identified in BC diagnostics to assess their relevance in CRC detection. Using data retrieved from The Human Protein Atlas [12], we analyzed the RNA expression levels of these markers in 63 CRC cell lines. We evaluated the cancer specificity of these markers by determining their expression in other types of cancers.

These results are summarized in Figure 1, where the bar graph illustrates the RNA expression levels of the 10 TACT markers in the CRC cell lines. *EPCAM* and *NPTN* exhibited significantly elevated expression levels in CRC cell lines, with *EPCAM* showing the highest expression among all the markers. The high expression levels suggest that these two markers are particularly relevant for CRC detection. Markers, including *KRT19*, *MKI67*, and *VIM*, were also highly expressed. In contrast, the markers *ERBB2*, *TERT*, and *SNAI2* were undetectable in CRC cell lines, suggesting they may not be useful for CRC diagnosis.

### 2.2. TACT Marker Expression in CRC Cell Lines vs. Immune Cells

As this study aimed to develop a blood-based diagnostic assay, ensuring that the selected markers were minimally expressed in immune cells was crucial, as high expression in immune cells could interfere with assay specificity and introduce background noise. To this end, we compared the RNA expression levels of the 10 TACT markers in CRC cell lines with their expression in immune cells using data from The Human Protein Atlas [9]. Figure 1 shows a comparison of the RNA expression levels between CRC cell lines and immune cells, identifying the markers more likely to be specific to CRC and less likely to cause background interference.

### 2.3. Comparative Analysis of TACT Marker Expression in Patients with CRC and Healthy Control Blood Samples

Following initial confirmation that not all 10 TACT markers were specifically associated with CRC, we next aimed to determine whether these markers could be detected at statistically significant levels in the blood of patients with CRC compared with healthy controls. Two hundred and six participants were recruited for this experiment: 107 patients with stage I–IV CRC and 99 healthy controls. Table 1 summarizes the demographic and clinical characteristics of participants in both the training and test cohorts. Data were categorized by age (under 50 and 50+ years), sex (male and female), and CRC stage, following TNM classification (Stages I–IV). This categorization provides an overview of the patient and control group distributions, serving as a baseline for evaluating the representativeness of the study cohort and the generalizability of the model results across age, sex, and disease stages.

To recruit a sufficient number of patients with CRC, the study utilized two cohorts from a gastroenterology department and a health screening center. Details of the study design are described in Table 1. In the gastroenterology department, blood samples were obtained from patients who had provided informed consent and were scheduled to undergo a colonoscopy. Colonoscopy was performed after blood collection, and patients with cancer confirmed by biopsy were included in the study. At the health screening center, informed consent was obtained from individuals undergoing routine health checkups. Blood samples were collected prior to colonoscopy procedures being performed. Patients without significant endoscopic findings were classified as healthy controls.

The results of TACT RNA expression analysis are shown in Figure 2.

Our findings indicated distinct expression patterns for TACT markers in CRC compared with those in BC, with specific markers (e.g., KRT19, FOXA2, and SNAIL2) showing significant upregulation in CRC, whereas others (e.g., EPCAM, ERBB2, and MKI67) were downregulated. This suggests cancer-type specificity in the RNA marker profiles, underscoring the need for tailored panels. Statistically significant differences were confirmed using *p*-values and 95% confidence intervals, which enhanced the reliability of these findings for clinical applications.

Several markers exhibited decreased expression in the blood of patients with CRC compared with that in HCs. Markers such as *EPCAM*, *ERBB2*, *MKI67*, *MCAM*, and *NPTN* showed statistically significant reductions in their expression levels. This was particularly unexpected, given their established association with cancer proliferation and EMT processes in other cancer types, such as BC. The expression levels of the 10 TACTs in CRC and BC samples are summarized in Table 2. TACT markers were categorized based on their biological roles as epithelial (EPCAM and KRT19), proliferation (ERBB2, TERT, and MKI67), or EMT (MCAM, VIM, FOXA2, SNAI2, and NPTN) markers. For comparison, findings from the previous study on BC were included [5]. The marker expression profiles in CRC vs. BC suggest cancer-type specificity with distinct expression patterns, emphasizing the need for tailored diagnostic markers for each cancer type.

Notably, *VIM* and *TERT*, which have shown diagnostic potential in BC studies, did not exhibit statistically significant differences in expression between the CRC and healthy control groups, indicating a divergence in the behavior of these markers between different cancer types.

These results highlighted the variability of TACT marker expression in different cancers. Although these markers hold diagnostic promise for BC, their expression profiles in CRC suggest that a tailored approach is necessary for CRC diagnostics. The varying expression levels across the 10 TACT markers indicate that CRC-specific panels must be developed to achieve high sensitivity and specificity for early CRC detection. Consequently, we selected eight statistically significant TACTs—*EPCAM*, *KRT19*, *ERBB2*, *MKI67*, *MCAM*, *FOXA2*, *SNAI2*, and *NPTN*—which we designated as colorectal TACTs (C-TACTs). Subsequently, these markers were used to develop an AI model to detect CRC.

### 2.4. Development of AI Models Using C-TACTs

The dataset of 206 blood samples (107 patients with CRC and 99 HCs) collected from the participants was randomly divided into training and test sets in a 7:3 ratio. The training dataset was used for model development, and the test dataset was used to validate the developed model and demonstrate its final performance (Figure 3). By inputting data from the training dataset, the model was trained to differentiate the CRC group from the control group, and model performance was confirmed using the area under the receiver operating characteristic curve (AUROC) and area under the precision–recall curve (AUPRC) utilizing independent data from the test dataset. The evaluated machine learning algorithms included GLM, DNN, RF, GBM, and AutoML.

Using the test set (35 patients with CRC and 22 HCs), the AUROC and AUPRC values for the different models were as follows: 0.912 and 0.929 for the DNN model; 0.841 and 0.847 for the AutoML model; 0.879 and 0.928 for the RF model; 0.891 and 0.930 for the GBM model; and 0.844 and 0.844 for the GLM model, respectively (Figure 4).

The performance metrics, particularly sensitivity and specificity, are summarized in Figure 5. Compared with the other four models, the DNN model exhibited superior accuracy (87.7%), sensitivity (85.7%), and specificity (90.9%).

Subsequently, we evaluated the sensitivity of the DNN models for each CRC stage. Our analysis revealed that the DNN model exhibited superior sensitivity in detecting stage I CRC cases (Figure 6). This finding suggests that the model is particularly effective in identifying early-stage CRC, potentially enhancing early detection and improving patient outcomes. Early detection at stage I is critical because it is associated with a significantly higher survival rate and more effective treatment options. Therefore, the high sensitivity of the DNN model in stage I cases underscores its potential utility as a screening tool, particularly for at-risk patients who may benefit from early interventions.

The sensitivity and specificity of the C-TACT DNN model were evaluated across various age and sex groups in the test cohort. The model exhibited elevated sensitivity across all groups (Figure 7). These results suggest that the C-TACT DNN model maintains high accuracy across diverse demographic groups, making it a promising tool for early, non-invasive CRC detection.

### 2.5. Analysis of Different Cancer Types Using a C-TACT DNN Model

To further assess the robustness of the prediction model, we applied the DNN-based model to BC and ovarian (OC) and cervical (CC) cancers, with eight samples for each cancer type. Owing to the absence of complete information for all 10 markers, we utilized only nine markers, excluding *MCAM*. To mitigate this constraint, we supplemented the absence of *MCAM* data by utilizing the expression profile of *MKI67*, which has a similar expression pattern to *MCAM* in CRC. The prediction outcomes based on the DNN model are presented in Figure 8.

The model demonstrated variable positive rates across these different cancer types, indicating its potential adaptability, but also highlighted some limitations in generalizability across distinct malignancies. However, the differences observed in specificity may reflect variations in the expression patterns of TACTs that are unique to each cancer type. These results imply that although the model can be applied broadly to different cancers, additional optimization and marker adjustments may be necessary to enhance the diagnostic precision for each specific cancer type. Adaptation of the DNN model using selected markers tailored to individual cancers could further improve performance, allowing for more accurate and reliable non-invasive screening across multiple cancer types.

Overall, these results suggest that the substitution strategy, although beneficial in some cases, may not be uniformly applicable to all cancer types. The effectiveness of each marker varies depending on cancer type, emphasizing the need for context-specific marker selection.

## 3. Discussion

This study highlights the promising utility of RNA-based liquid biopsies, specifically through a multi-marker panel of TACTs combined with machine learning, as a non-invasive diagnostic tool for CRC. Unlike traditional CRC diagnostic methods, such as colonoscopy or stool-based DNA tests, which often face limitations in early detection and patient compliance, RNA biomarkers offer unique advantages by capturing real-time gene expression. This dynamic approach provides insights into the ongoing biological processes, including tumor proliferation, immune evasion, and EMT, which are essential for understanding cancer progression [5,13]. In particular, the capacity to detect these processes in real time enhances early detection capabilities, which are critical for improving CRC outcomes.

A notable finding in this study was the cancer-type specificity observed among TACT markers, such as EPCAM and ERBB2, which displayed differential expression patterns between CRC and BC. These markers were significantly upregulated in BC, reflecting their roles in cell adhesion and proliferation, which are commonly associated with BC tumorigenesis [14]. However, their downregulation in CRC suggests an interaction with the unique CRC microenvironment, potentially influenced by factors such as leukocytosis, a common inflammatory condition in patients with CRC that may dilute tumor-derived diagnostic panels, as biomarkers can be differentially regulated depending on the cellular context and microenvironmental conditions of the tumor. This differentiation may be crucial for developing accurate diagnostic tools adaptable to various cancers, highlighting the value of further research to clarify the molecular mechanisms underlying these observed differences.

Our study also demonstrated the strengths of RNA-based diagnostics compared with cfDNA-based methods. Although cfDNA provides valuable insights into genetic mutations, it predominantly captures a static genomic profile that may overlook real-time changes in gene expression, which are critical for monitoring active tumor processes [15]. RNA markers, particularly those derived from circulating tumor cells and exosomes, allow for the dynamic monitoring of tumor biology, capturing shifts in gene expression that reflect the tumor’s immediate response to biological cues and treatments [16]. The stability of exosomal RNA, coupled with its ability to reflect the tumor microenvironment, makes RNA-based assays particularly advantageous for early cancer detection, where the timely identification of molecular changes is crucial [17].

Integrating multiple RNA markers instead of relying on a single biomarker further enhances diagnostic robustness. Previous BC studies have demonstrated that single biomarkers often lack sufficient sensitivity because they do not fully capture the biological complexity inherent in tumor development and metastasis [5]. By employing a multi-marker approach, this study addresses these limitations, with the DNN model achieving high sensitivity (85.7%) and specificity (90.9%) for CRC detection. The DNN model demonstrated the best diagnostic performance in our study, making it the optimal model for CRC detection. DNNs are machine learning algorithms that learn hierarchical patterns through multiple layers of interconnected neurons, allowing them to model complex nonlinear relationships in high-dimensional data [10,11]. Their key advantages include the ability to automatically extract meaningful features from raw data, scalability in handling large datasets, and the use of techniques such as dropout and batch normalization to prevent overfitting. DNNs have been widely applied in CRC research, particularly for analyzing histopathological images and genomic data, achieving high accuracy in detecting malignancies, and identifying biomarkers associated with prognosis and personalized treatment [18,19]. They also play a significant role in endoscopic image interpretation, aiding in diagnosing polyps and CRC lesions early. In our study, the DNN model outperformed the other models, demonstrating its ability to uncover complex patterns and deliver reliable predictions for blood-based CRC diagnostics.

The performance of the established DNN model rivals established methods such as stool-based DNA tests (e.g., Cologuard; Exact Sciences Corporation, Madison, WI, USA) and emphasizes the clinical potential of our approach [20]. Additionally, the ability of the model to maintain high accuracy across different age groups supports its potential utility in both routine screening and early detection efforts for at-risk populations that might otherwise avoid invasive procedures. We demonstrated the high sensitivity of the AI models across CRC stages I–IV (Figure 6), emphasizing their potential for early detection. Additionally, to evaluate the cross-cancer applicability of the C-TACT markers, we applied the DNN model to breast (BC), ovarian (OC), and cervical (CC) cancer samples (Figure 8). The model achieved positive prediction rates of 88% (BC), 75% (CC), and 0% (OC), highlighting the adaptability of the TACT markers while identifying limitations in cross-cancer specificity. However, the specificity of TACT markers across CRC subtypes and their potential cross-reactivity with non-malignant gastrointestinal diseases was not investigated. Future studies will incorporate subtype-specific analyses and additional cohorts with non-malignant diseases to strengthen the diagnostic utility of TACT markers.

Many TACT markers used in the present study, such as EPCAM and MKI67, have been implicated in fundamental biological processes relevant to various cancer types, including cell proliferation and EMT. Therefore, this study suggests that the RNA panel may be applicable beyond CRC, potentially serving as a foundation for multicancer diagnostic platforms. Despite some markers, such as EPCAM, being broad-spectrum, their unique expression patterns, combined with other TACT markers, formed a CRC-specific diagnostic panel. While the same 10 markers were previously applied in BC research, distinct expression profiles enabled the construction of cancer-specific panels. For CRC, 8 markers were selected for their significant differential expression, optimizing the diagnostic panel’s specificity and sensitivity. However, the adaptability of this approach to other cancers, such as cervical or ovarian cancer, warrants further investigation, particularly to understand how these markers might function differently depending on cancer type and stage. Validation studies on diverse cancer types could broaden the applicability of this diagnostic model and underscore its utility as a comprehensive screening tool for multiple cancers.

Despite these promising findings, certain limitations of this study must be acknowledged. The relatively small sample size in this study could have affected the robustness of our results and limited the generalizability of our findings to a broader CRC population. We acknowledge that the data used in this study were collected from specific healthcare institutions, which may limit the generalizability of our findings. Future studies will include participants from multiple regions and diverse populations to improve data representativeness and reduce selection bias. Larger and more diverse cohorts are necessary to confirm these results and ensure diagnostic accuracy across different demographic and clinical subgroups, including patients with varying cancer stages and comorbidities. Additionally, our TACT marker selection was based on previous studies on BC, which may not fully capture CRC-specific biological mechanisms. Future studies should explore CRC-specific markers, investigate the mechanistic roles of each marker in CRC progression, and enhance the scientific rigor of marker selection.

Moreover, while we observed that markers such as EPCAM and ERBB2 were downregulated in CRC, further research is required to clarify the impact of inflammatory responses, such as leukocytosis, on the detectability of circulating RNA in patients with CRC. This line of inquiry could provide insight into the reliability of certain RNA markers under specific clinical conditions, potentially leading to improved diagnostic accuracy for CRC and other inflammation-associated cancers.

This study primarily focused on evaluating the diagnostic potential of 10 tumor-associated circulating transcript (TACT) markers for CRC detection using a liquid biopsy approach. While these markers were selected based on their biological relevance as circulating RNA biomarkers, we acknowledge that a more comprehensive approach incorporating genomic data, including cancer-driving mutations, gene rearrangements, and pathogenic isoform variants, would provide deeper insights into CRC biology. Future research will explore the integration of transcriptomic and genomic data, enabling a more holistic diagnostic framework. Additionally, expanding the marker panel to include established CRC driver genes and utilizing larger, multi-institutional datasets will further enhance the robustness and applicability of the diagnostic model.

In summary, this study established the feasibility of multi-marker RNA-based liquid biopsy, powered by machine learning, as a potential non-invasive diagnostic tool for CRC. Future studies expanding the model’s validation to larger and more diverse patient populations, refining marker selection, and testing its applicability across multiple cancer types could pave the way for a robust and versatile RNA-based diagnostic platform with substantial implications for personalized cancer screening and management.

## 4. Materials and Methods

### 4.1. Patient Cohorts

Whole blood samples were collected from 107 patients with CRC and 99 healthy controls (HC) recruited from the Department of Gastroenterology at Severance Hospital, Gangnam Severance Hospital, and Gangbuk Samsung Hospital, Seoul, Republic of Korea (IRB #4-2017-0148, #3-2017-0024). Healthy controls were recruited through routine health screening at Wonju Severance Christian Hospital (IRB #CR319115). The inclusion criteria for patients included histological confirmation of CRC based on colonoscopy and histological results with dysplasia grade level, villous component protein, and size and number of polyps, according to the European Society of Gastrointestinal Endoscopy (ESGE). The exclusion criteria included prior CRC resection or evidence of hereditary colorectal cancer syndrome. The staging criteria for patients with CRC from stages I to IV followed the guidelines set forth by the ESGE. Among the 107 blood samples collected from patients with CRC, 44 were classified as stage I, 18 as stage II, 23 as stage III, and 22 as stage IV. This classification was based on the widely accepted TNM staging system, ensuring consistency with the ESGE standards for accurate patient stratification and treatment planning. The HC group included individuals with no significant findings after colonoscopy and no other cancers. Informed consent was obtained from all the participants. The clinicopathologic characteristics of the participants are shown in Table 3.

### 4.2. Blood Collection

Blood samples were collected by venipuncture using a 3 mL Tempus™ Blood RNA Tube (Thermo Fisher Scientific, Waltham, MA, USA) as secondary or subsequent order blood collection to avert the entry of epithelial cells into the blood. Following blood collection, the tube was vortexed for 10 s to ensure complete mixing of the blood with 6 mL stabilizing reagent contained in the tube. Subsequently, the Tempus Blood RNA Tube was maintained in an upright position at room temperature (18 to 25 °C) for no longer than 5 days before processing or moving to a refrigerator (2 °C to 8 °C) or freezer (−20 °C).

### 4.3. RNA Isolation and cDNA Synthesis

RNA was extracted using a Tempus Spin RNA Isolation Kit (Thermo Fisher Scientific) following the manufacturer’s protocol. The quality of the isolated RNA was assessed using an RNA 6000 Nano LabChip with an Agilent 2100 Bioanalyzer (Agilent Technologies, Santa Clara, CA, USA) and a NanoDrop spectrophotometer (Thermo Fisher Scientific).

Complementary DNA (cDNA) was synthesized using a High-Capacity cDNA Reverse Transcription Kit (Thermo Fisher Scientific). RNA calculated based on the measured concentration was diluted with nuclease-free water to achieve a concentration of 2 μg/14.2 μL. A mixture was prepared using 10× RT buffer, 25× deoxynucleotide triphosphate (dNTP) mix, 10× RT random primers, and MultiScribe™ Reverse Transcriptase (Invitrogen, Carlsbad, CA, USA) at a ratio of 10:4:10:5. All reagents were contained in a High-Capacity cDNA Reverse Transcription Kit (Applied Biosystems, Foster City, CA, USA). The total volume of the reaction mixture was modified according to the number of samples, and 5.8 μL of the mixture was distributed into each sample. Each sample was mixed thoroughly and briefly centrifuged to ensure even distribution.

Subsequently, cDNA was synthesized using random hexamers and dNTPs in a thermal cycler (Bio-Rad Laboratories, Hercules, CA, USA). The reaction was initiated at 25 °C for 10 min to allow priming and enzyme activation (Step 1), followed by incubation at 37 °C for 50 min to facilitate reverse transcription (Step 2). Subsequently, the reaction mixture was heated to 85 °C for 5 min to inactivate the reverse transcriptase enzyme (Step 3). All steps were conducted in accordance with the manufacturer’s instructions.

### 4.4. Quantitative PCR (qPCR)

The amplification of each gene was quantitatively measured using TaqMan^®^ Array CRC (TAC) Cards on a QuantStudio™ 7 pro-Real-Time PCR System (Thermo Fisher Scientific). qPCR was performed using 50 μL TaqMan^®^ Advanced Master Mix (Thermo Fisher Scientific) and 50 μL template cDNA to bring the final volume to 100 μL. Each reaction was conducted in duplicate using the same TAC. The TaqMan Fast Advanced Master Mix contained AmpliTaq Fast DNA Polymerase, uracil-*N*-glycosylase (UNG), dNTPs with dUTP, and ROX dye (passive reference). Gene expression levels were determined by normalization to the internal housekeeping gene *GAPDH*.

The relative gene expression was assessed using the comparative Ct (ΔΔCt) method [21]. The amount of target, normalized to an internal housekeeping gene and relative to a calibrator, was given by 2^−ΔΔCt^, which was then normalized according to Equation (1):ΔΔCt = [ΔCt(test) = Ct(target test) − Ct(reference test)] − [ΔCt(calibrator)(1)= Ct(target calibrator) − Ct(reference calibrator)]

### 4.5. Machine Learning Models

We developed and evaluated five machine learning models—GLM, RF, GBM, DNN, and AutoML—using TACT marker data to identify the optimal approach for CRC detection. To enhance model robustness and minimize overfitting, several strategies were employed. First, 5-fold cross-validation was conducted during training to evaluate performance across multiple data splits, ensuring generalizability within the dataset. Second, the dataset of 206 blood samples (107 CRC patients and 99 healthy controls) was randomly divided into training (70%) and independent test sets (30%). The test set, which was not involved in training or cross-validation, was reserved for final validation to assess generalizability to unseen data. The model parameters were tuned to optimize sensitivity and specificity, ensuring a balanced diagnostic performance. To assess the predictive accuracy of each model, ROC and PR curves were analyzed alongside the AUC metrics for a comprehensive evaluation.

Each model was processed in two sets: training and testing. For the training and validation of the model using artificial intelligence analysis, 70% of the total subjects, including 72 patients with CRC and 77 healthy volunteers, were used (the training set), and the model was created. The remaining 30% of the total population, including 35 patients with CRC and 22 healthy volunteers, were used to test the model.

### 4.6. Statistical Analysis

Two-group comparisons were performed using Student’s *t*-test in GraphPad Prism software (version 9.0; GraphPad Software, San Diego, CA, USA). Statistical significance was set at *p* < 0.05. ROC and PR curves were generated using the gglot2 package in R software (version 4.4.2, R Foundation for Statistical Computing, Vienna, Austria).

## 5. Conclusions

This study demonstrated the potential of a multi-marker RNA-based liquid biopsy approach combined with machine learning as a promising non-invasive diagnostic tool for CRC. By leveraging TACTs, our DNN model achieved high sensitivity and specificity, suggesting that this approach can significantly enhance early CRC detection and improve patient outcomes by addressing the compliance limitations associated with invasive methods. Furthermore, the ability of RNA-based diagnostics to capture real-time gene expression underscores their advantage over static cfDNA mutation analysis, providing a dynamic assessment of tumor biology.

However, to establish broader clinical utility, future studies with larger and more diverse cohorts are essential to confirm the generalizability of this approach. Additionally, refining marker selection to target CRC-specific pathways and validating this RNA panel across other cancer types may pave the way for a multicancer diagnostic platform. By expanding these investigations, RNA-based diagnostics hold promise as versatile tools for personalized cancer screening and monitoring, potentially transforming early detection strategies for various malignancies.

## Figures and Tables

**Figure 1 ijms-26-01477-f001:**
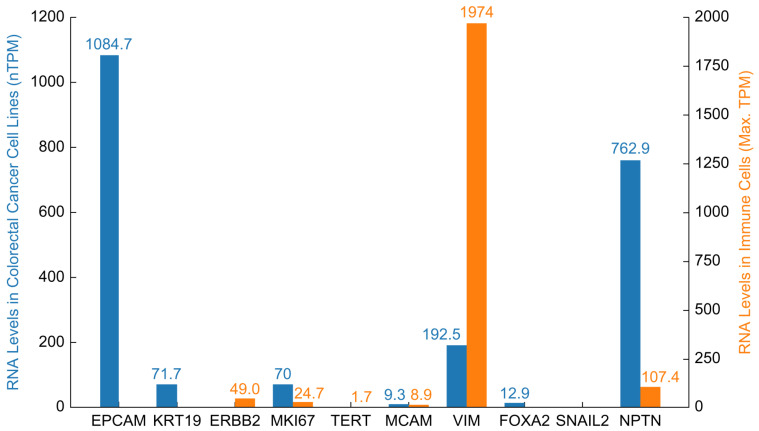
Comparison of RNA levels of tumor-associated circulating transcripts (TACT) markers in CRC cell lines and immune cells. The bar chart shows the RNA expression levels (in normalized transcripts per million, nTPM) of 10 TACT markers across 63 colorectal cancer cell lines (blue bars) compared with their maximum expression levels in immune cells (Max. TPM) (orange bars).

**Figure 2 ijms-26-01477-f002:**
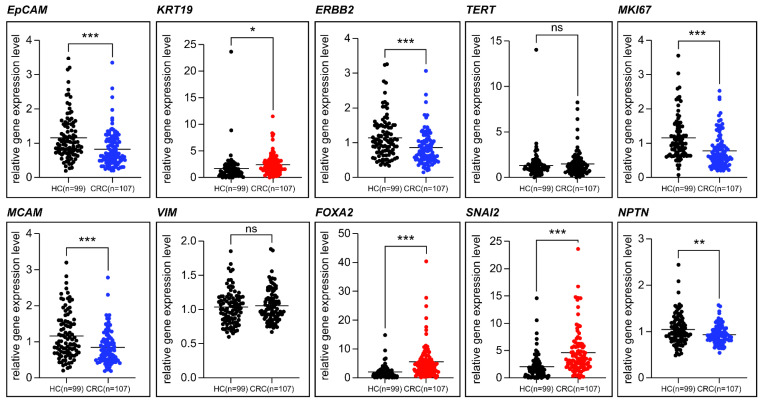
Expression levels of TACT markers in healthy controls and patients with colorectal cancer (CRC). The scatter plots compare the expression levels of 10 TACT markers—EPCAM, KRT19, ERBB2, TERT, MKI67, MCAM, VIM, FOXA2, SNAI2, and NPTN—between healthy controls (HCs) and patients with CRC. Each data point represents an individual sample, with statistical significance evaluated using *t*-tests. Significance levels are visually indicated, with *p*-values marked as follows: *p* < 0.05 (*), *p* < 0.01 (**), and *p* < 0.001 (***). The distinct expression profiles of TACT markers in CRC are highlighted, supporting their potential as diagnostic indicators. ns: no significance.

**Figure 3 ijms-26-01477-f003:**
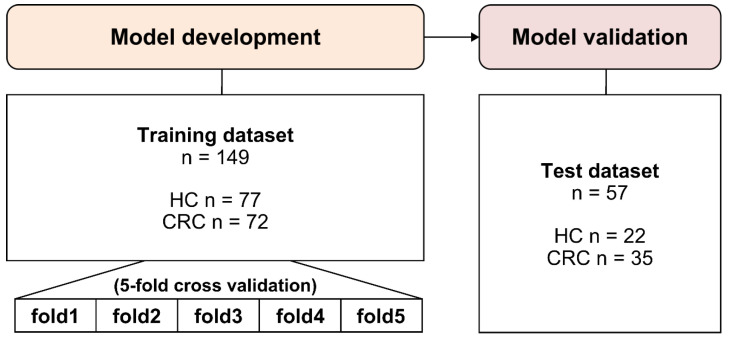
AI model development and validation process. The flowchart depicts the process of dataset division and model validation. The full dataset was split into training (70%) and test (30%) sets to build and validate the machine learning models, respectively. Cross-validation was performed within the training set to enhance model robustness. The rigorous approach used to ensure the reliability of the AI model’s performance is illustrated, emphasizing the methodological rigor in validating the diagnostic utility of the tumor-associated circulating transcript marker panel. HC, healthy controls; CRC, colorectal cancer.

**Figure 4 ijms-26-01477-f004:**
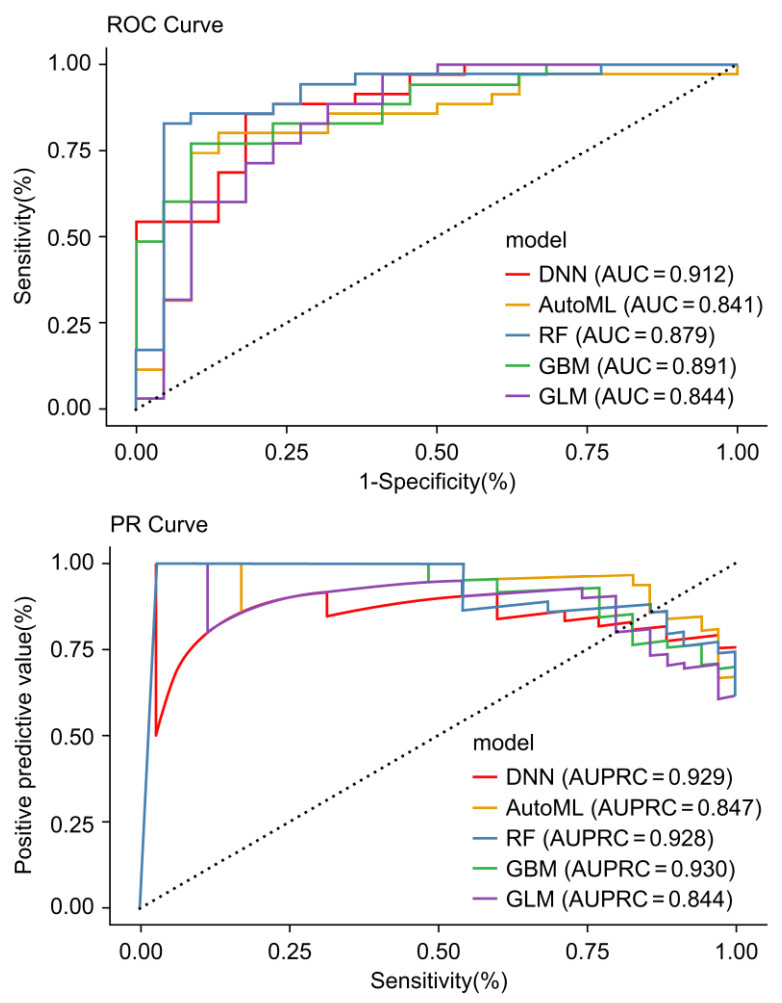
Receiver operating characteristic (ROC) and precision–recall (PR) curves for machine learning model performance. The composite plot displays ROC and PR curves for five machine learning models—deep neural network (DNN), automated machine learning algorithm detecting method (AutoML), random forest (RF), gradient boosting machine (GBM), and generalized linear model (GLM). The curves offer a visual assessment of each model’s diagnostic performance, with the area under each curve (AUC) quantitatively reflecting the model’s ability to distinguish between colorectal cancer and control samples. The ROC curves focus on sensitivity vs. 1-specificity, while the PR curves capture precision vs. recall, providing insights into the models’ effectiveness in a diagnostic context.

**Figure 5 ijms-26-01477-f005:**
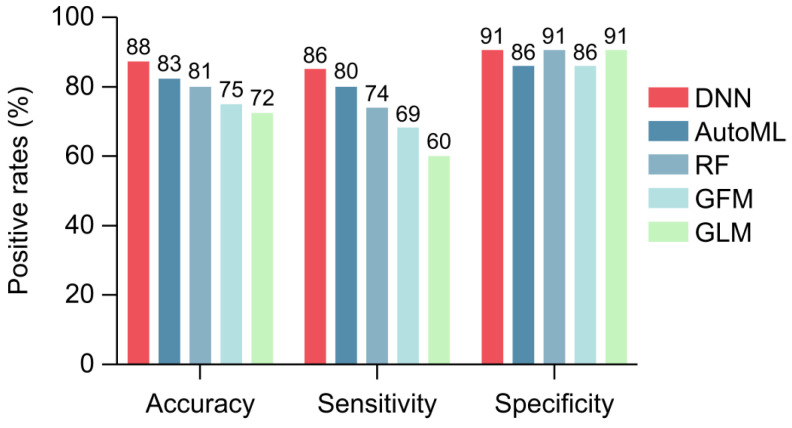
Comparison of model accuracy, sensitivity, and specificity. The bar chart illustrates the accuracy, sensitivity, and specificity of the five machine learning models—deep neural network (DNN), automated machine learning algorithm detecting method (AutoML), random forest (RF), gradient boosting machine (GBM), and generalized linear model (GLM)—in diagnosing colorectal cancer. Each model’s performance is color coded across these metrics, highlighting the DNN model’s superior accuracy (87.7%), sensitivity (85.7%), and specificity (90.9%). This chart underscores the DNN model’s robustness, establishing it as the most promising approach among the evaluated colorectal cancer (CRC) detection models.

**Figure 6 ijms-26-01477-f006:**
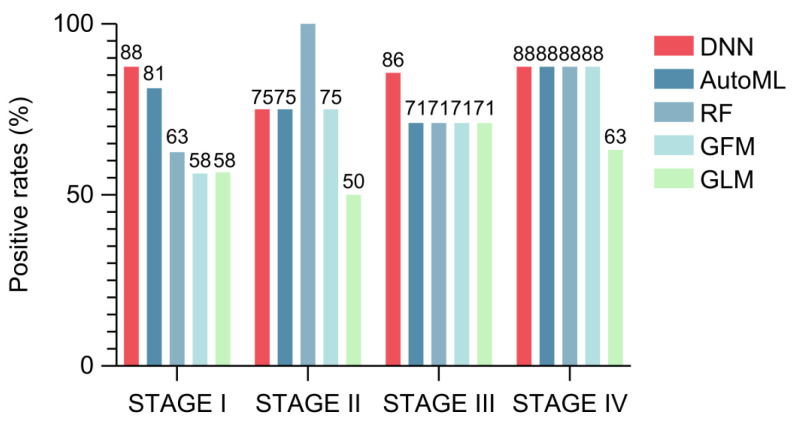
Sensitivity of AI models across colorectal cancer (CRC) stages. The bar chart shows the sensitivity of the AI models in detecting CRC across stages I–IV, based on C-TACT marker data. This model’s high sensitivity in early-stage CRC cases demonstrates its potential utility for early detection, where timely intervention can greatly impact patient outcomes. The figure illustrates the diagnostic reach of the deep neural network (DNN) model across cancer stages, emphasizing its applicability in detecting CRC from the earliest stages.

**Figure 7 ijms-26-01477-f007:**
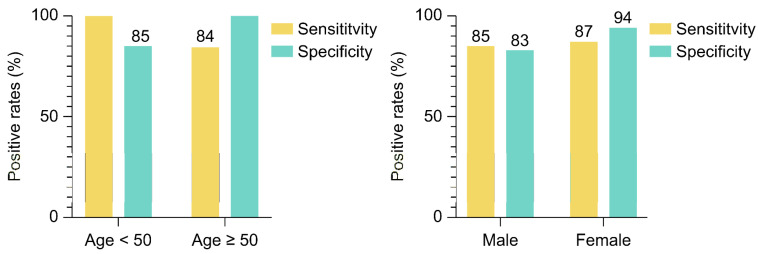
Sensitivity and specificity of C-TACT DNN model across age and gender. The figure presents the deep neural network (DNN) model’s diagnostic performance across different demographic subgroups in the test cohort. Sensitivity and specificity are shown for participants under 50 and those 50 and older, as well as across male and female groups. These results indicate that the model maintains high accuracy across diverse demographic groups, reinforcing its suitability for broad application in colorectal cancer screening.

**Figure 8 ijms-26-01477-f008:**
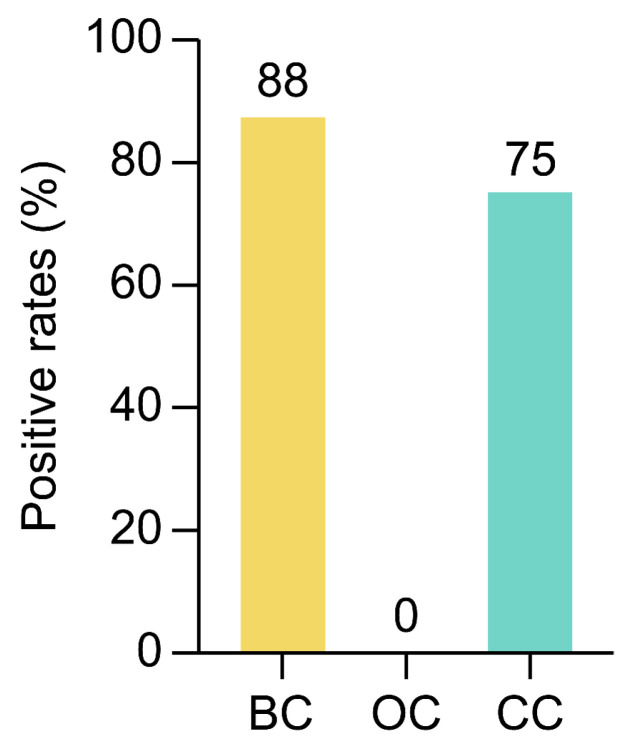
Cross-cancer prediction using the C-TACT DNN model. The figure illustrates the positive prediction rates when applying the deep neural network (DNN) model to samples from different cancer types (breast, ovarian, and cervical), each represented by eight samples. While the model shows variable positive rates, this analysis highlights the adaptability of the TACT markers across malignancies while also identifying limitations in cross-cancer specificity. This analysis suggests potential avenues for expanding the model’s diagnostic scope, with marker adjustments to enhance cancer-type specificity.

**Table 1 ijms-26-01477-t001:** Clinicopathologic characteristics of patients with colorectal cancer (CRC) and healthy controls.

Cohort	Training Cohort, *n* (%)		Test Cohort, *n* (%)	
	Healthy Control (*n* = 77)	CRC (*n* = 72)	Healthy Control (*n* = 22)	CRC (*n* = 35)
Age				
<50	41 (53.0)	8 (11.0)	13 (59.0)	4 (11.5)
≥50	36 (47.0)	64 (89.0)	9 (41.0)	31 (88.5)
Sex				
Male	38 (50.0)	41 (57.0)	6 (27.0)	20 (57.0)
Female	39 (50.0)	31 (43.0)	16 (73.0)	15 (43.0)
CRC Stage				
I		28 (39.0)		16 (46.0)
II		14 (19.5)		4 (11.0)
III		16 (22.0)		7 (20.0)
IV		14 (19.5)		8 (23.0)

**Table 2 ijms-26-01477-t002:** Comparative analysis of TACT marker expression in breast cancer (BC) and colorectal cancer (CRC).

Group	TACT	BC		CRC	
		Difference Between Means (BC–HC)	*p* Value	Difference Between Means (CRC–HC)	*p* Value
Epithelial markers	*EPCAM*	7.1	***	−0.3	***
	*KRT19*	12.3	***	0.7	*
Proliferation markers	*ERBB2*	0.8	**	−0.3	***
	*TERT*	1.5	**	0.1	ns
	*MKI67*	1.5	***	−0.4	***
Epithelial-to-mesenchymal markers	*MCAM*	5.3	*	−0.3	***
	*VIM*	0.2	*	0.02	ns
	*FOXA2*	3.9	**	3.7	***
	*SNAI2*	3.4	**	2.6	***
	*NPTN*	0.2	*	−0.1	**

Statistically significant differences in expression are indicated by asterisks: *p* < 0.05 (*), *p* < 0.01 (**), *p* < 0.001 (***), and ns for non-significant.

**Table 3 ijms-26-01477-t003:** Clinicopathologic characteristics of study participants.

Cohort	Healthy Control, *n* (%)	Colorectal Cancer, *n* (%)
Age		
<50	54 (82.0)	12 (18.0)
≥50	45 (32.0)	95 (68.0)
Sex		
Male	44 (42.0)	61 (58.0)
Female	55 (54.5)	46 (45.5)
CRC Stage		
I		44 (41.0)
II		18 (16.5)
III		23 (21.5)
IV		22 (21.0)

CRC, colorectal cancer.

## Data Availability

Data is contained within the article.

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
