# Peer review of "Machine Learning-Enabled Non-Invasive Screening of Tumor-Associated Circulating Transcripts for Early Detection of Colorectal Cancer"

_ijms, 2025, doi:10.3390/ijms26041477_

Round 1
Reviewer 1 Report
Comments and Suggestions for Authors
The manuscript by Jin Han et al. advocates for using tumor-associated circulating transcripts (TACTs) alongside machine learning for non-invasive colorectal cancer (CRC) screening. The study is well-structured, with clear objectives and a robust methodology integrating biological and computational strategies. The machine learning approach demonstrated effective analysis and recognition capabilities, suggesting that the manuscript could be published after minor revisions.
Comments:
1. Please clarify how the algorithms differentiate between various tumor models, considering that some tumor markers, like EpCAM, are broad-spectrum.
2. Given the use of multiple machine learning models, there's a risk of overfitting, where performance may excel on training data but falter on new, unseen data due to flexibility in model selection and parameter tuning.
3. The case studies were sourced from specific healthcare institutions, potentially limiting the representativeness of the data and introducing selection bias concerning a broader, diverse population.
4. While the article highlights the high sensitivity and specificity of TACT markers in CRC detection, it lacks detailed discussion of the specificity of these markers across different cancer stages and subtypes and the influence of other disease states.
Author Response
[Comments 1] Please clarify how the algorithms differentiate between various tumor models, considering that some tumor markers, like EpCAM, are broad-spectrum.
[Response 1]
Thank you for your insightful question regarding how our machine learning models differentiate between tumor models, considering that some markers, like EpCAM, are widely expressed across multiple cancer types.
In our study, we utilized Tumor-Associated Circulating Transcripts (TACTs), which are RNA-based biomarkers, rather than solely relying on protein markers like EpCAM. TACTs provide a more dynamic and functional representation of tumor biology, including tumor-specific gene expression changes and immune interactions. Unlike EpCAM, which is a broadly expressed surface protein, TACTs capture tumor- and stage-specific transcriptional activity, allowing for more refined classification.
To enhance the specificity of our models, we incorporated machine learning-driven feature selection to identify the most informative transcriptomic markers. Our Deep Neural Network (DNN) model further optimizes marker weighting, ensuring that commonly expressed genes like EpCAM do not overly influence tumor differentiation. Additionally, the integration of multiple transcriptomic features rather than single broad-spectrum markers allows the algorithm to recognize distinct expression patterns unique to colorectal cancer (CRC).
By leveraging AI-based analysis, our approach minimizes misclassification due to broadly expressed markers and prioritizes tumor-specific transcriptomic signatures. This strategy enhances diagnostic accuracy and improves early-stage CRC detection, as demonstrated by our model’s high sensitivity (85.7%) and specificity (90.9%).
We have clarified these points in the revised manuscript (Discussion, line 348-353).
[Revised manuscript]
“Many TACT markers used in the present study, such as EPCAM and MKI67, have been implicated in fundamental biological processes relevant to various cancer types, including cell proliferation and EMT. Therefore, this study suggests that the RNA panel may be applicable beyond CRC, potentially serving as a foundation for multicancer diagnostic platforms. Despite some markers, such as EPCAM, being broad-spectrum, their unique expression patterns, combined with other TACT markers, formed a CRC-specific diagnostic panel. While the same 10 markers were previously applied in BC research, distinct expression profiles enabled the construction of cancer-specific panels. For CRC, 8 markers were selected for their significant differential expression, optimizing the diagnostic panel’s specificity and sensitivity. However, the adaptability of this approach to other cancers, such as cervical or ovarian cancer, warrants further investigation, par-ticularly to understand how these markers might function differently depending on cancer type and stage. Validation studies on diverse cancer types could broaden the applicability of this diagnostic model and underscore its utility as a comprehensive screening tool for multiple cancers.”
[Comments 2] Given the use of multiple machine learning models, there's a risk of overfitting, where performance may excel on training data but falter on new, unseen data due to flexibility in model selection and parameter tuning.
[Response 2]
Thank you for raising this important concern regarding the potential risk of overfitting when using multiple machine learning models. This is a valid consideration, particularly given the flexibility in model selection and parameter tuning.
To mitigate overfitting, we employed multiple strategies. First, we conducted 5-fold cross-validation during model training to assess performance across different data splits. This approach ensured that the models generalized well to unseen data within the dataset, reducing the risk of overfitting. Second, the dataset, comprising 206 blood samples (107 CRC patients and 99 healthy controls), was randomly divided into training (70%) and independent test sets (30%). The test set, which was not involved in model training or cross-validation, was used solely for final validation to evaluate generalizability. Consistent performance metrics observed on the test set demonstrated the robustness of the models. Lastly, by focusing on a curated panel of TACT markers with significant differential expression between CRC and HC samples, we minimized the inclusion of noisy or irrelevant features, further reducing the risk of overfitting.
We have elaborated on these measures in the revised manuscript (Method 4.5, lines 462–470).
[Revised manuscript]
“We developed and evaluated five machine learning models—GLM, RF, GBM, DNN, and AutoML—using TACT marker data to identify the optimal approach for CRC de-tection. To enhance model robustness and minimize overfitting, several strategies were employed. First, 5-fold cross-validation was conducted during training to evaluate per-formance across multiple data splits, ensuring generalizability within the dataset. Second, the dataset of 206 blood samples (107 CRC patients and 99 healthy controls) was randomly divided into training (70%) and independent test sets (30%). The test set, which was not involved in training or cross-validation, was reserved for final validation to assess gen-eralizability to unseen data.”
[Comments 3] The case studies were sourced from specific healthcare institutions, potentially limiting the representativeness of the data and introducing selection bias concerning a broader, diverse population.
[Response 3]
Thank you for highlighting this important concern regarding the generalizability of our findings. We acknowledge that the data used in this study were collected from specific healthcare institutions, which may introduce potential selection bias and limit applicability to broader and more diverse populations. To mitigate this limitation, we recruited participants from four hospitals: Severance Hospital, Gangnam Severance Hospital, and Gangbuk Samsung Hospital in Seoul, as well as Wonju Severance Christian Hospital. These participants were drawn from various departments, including Gastroenterology, Family Medicine, and Health Screening Centers, ensuring a diverse representation of clinical backgrounds. Additionally, our co-authors, specialists from fields such as Gastroenterology, Family Medicine, and Microbiology, contributed to patient recruitment and study design, further enhancing the study's rigor.
Despite these efforts, we recognize the need for larger, multi-institutional datasets to improve representativeness. Future studies will expand sample collection to include participants from diverse regions and demographics, addressing selection bias and enhancing data robustness. External validation using publicly available datasets or independent cohorts will also be conducted to further confirm the model's generalizability.
These revisions have been included in the revised manuscript (Discussion, lines 362–365) to strengthen the study's rigor and transparency.
[Revised manuscript]
“Despite these promising findings, certain limitations of this study must be acknowledged. The relatively small sample size in this study could have affected the robustness of our results and limited the generalizability of our findings to a broader CRC population. We acknowledge that the data used in this study were collected from specific healthcare institutions, which may limit the generalizability of our findings. Future studies will include participants from multiple regions and diverse populations to im-prove data representativeness and reduce selection bias. Larger and more diverse cohorts are necessary to confirm these results and ensure diagnostic accuracy across different demographic and clinical subgroups, including patients with varying cancer stages and comorbidities. Additionally, our TACT marker selection was based on previous studies on BC, which may not fully capture CRC-specific biological mechanisms. Future studies should explore CRC-specific markers, investigate the mechanistic roles of each marker in CRC progression, and enhance the scientific rigor of marker selection.”
[Comments 4] While the article highlights the high sensitivity and specificity of TACT markers in CRC detection, it lacks detailed discussion of the specificity of these markers across different cancer stages and subtypes and the influence of other disease states.
[Response 4]
Thank you for your insightful comment regarding the specificity of TACT markers across different cancer stages, subtypes, and potential cross-reactivity with other disease states.
To address marker performance across cancer stages, we analyzed the sensitivity of our AI models in detecting CRC across stages I–IV, as shown in Figure 6. The results demonstrate that the deep neural network (DNN) model achieves consistently high sensitivity across all stages, with particularly strong performance in early-stage CRC (Stage I: 88%) and advanced stages (Stage IV: 88%). These findings underscore the potential of the model for both early detection and advanced-stage CRC, which are critical for improving patient outcomes.
Regarding cross-cancer predictions, we further assessed the adaptability of the DNN model by applying it to samples from breast cancer (BC), ovarian cancer (OC), and cervical cancer (CC) (Figure 8). Positive prediction rates varied across cancer types, with 88% for BC, 75% for CC, and 0% for OC. These results highlight the flexibility of TACT markers in different malignancies while also revealing limitations in cross-cancer specificity. Refining marker panels to enhance specificity for different cancer types will be essential to expanding the diagnostic scope of the model.
While our study did not directly investigate CRC subtypes or non-malignant gastrointestinal diseases, we recognize that such factors may influence marker performance. The healthy control group in this study excluded individuals with known inflammatory or gastrointestinal conditions, but future studies will incorporate additional cohorts, including CRC subtypes (e.g., based on molecular or clinical classifications) and non-malignant diseases, to further evaluate marker specificity and diagnostic utility.
We have expanded the Discussion section (lines 333–343) to address these limitations and outline areas for future research.
[Revised manuscript]
“The performance of the established DNN model rivals established methods such as stool-based DNA tests (e.g., Cologuard; Exact Sciences Corporation, Madison, WI, USA) and emphasizes the clinical potential of our approach [20]. Additionally, the ability of the model to maintain high accuracy across different age groups supports its potential utility in both routine screening and early detection efforts for at-risk populations that might otherwise avoid invasive procedures. We demonstrated high sensitivity of the AI models across CRC stages I–IV (Figure 6), emphasizing their potential for early detection. Additionally, to evaluate the cross-cancer applicability of the C-TACT markers, we applied the DNN model to breast (BC), ovarian (OC), and cervical (CC) cancer samples (Figure 8). The model achieved positive prediction rates of 88% (BC), 75% (CC), and 0% (OC), highlighting the adaptability of the TACT markers while identifying limitations in cross-cancer specificity. However, the specificity of TACT markers across CRC subtypes and their potential cross-reactivity with non-malignant gastrointestinal diseases was not investigated. Future studies will incorporate subtype-specific analyses and additional cohorts with non-malignant diseases to strengthen the diagnostic utility of TACT markers.”
Reviewer 2 Report
Comments and Suggestions for Authors
The authors themselves stated the shortcomings of their own work. The small number of samples analyzed, as well as the selection of TACT markers (which the authors used in their previously published work on breast cancer), which do not fully correspond to the biological nature of CRC. What else can I add?
Author Response
[Comments] The authors themselves stated the shortcomings of their own work. The small number of samples analyzed, as well as the selection of TACT markers (which the authors used in their previously published work on breast cancer), which do not fully correspond to the biological nature of CRC. What else can I add?
[Response]
Thank you for your thoughtful comments. We acknowledge the limitations of our study, including the relatively small sample size and the selection of tumor-associated circulating transcript (TACT) markers previously applied in breast cancer (BC) research, and we appreciate the opportunity to elaborate further on these points.
This study is part of a broader investigation into the utility of Tumor-Associated Circulating Transcripts (TACTs) as minimally invasive biomarkers for cancer detection. While TACT markers were originally validated in BC diagnostics, this study explored their potential application in colorectal cancer (CRC). Notably, our initial analysis revealed distinct expression profiles of these markers between BC and CRC, despite using the same set of markers. This finding underscores the flexibility of TACT markers, which, when analyzed through machine learning algorithms, enable the development of cancer-specific diagnostic panels. In this study, the expression patterns of TACT markers were integrated to construct a CRC-specific diagnostic panel, with eight of the ten markers demonstrating significant differential expression between CRC and healthy controls, highlighting their diagnostic relevance. These results suggest that TACT markers may have broader diagnostic utility across multiple cancer types. Future research will expand the exploration of TACT markers to refine their specificity and adaptability for CRC and other malignancies.
We also recognize that the sample size of 107 CRC patients and 99 healthy controls limits the statistical power and generalizability of our findings. To strengthen these findings, future research will focus on expanding the dataset through multicenter collaborations and including broader, more diverse populations. Such efforts will improve the robustness and applicability of the diagnostic model.
Despite these limitations, this study advances the field of liquid biopsy by demonstrating the adaptability and scalability of TACT markers in CRC diagnostics. The flexible framework established here provides a strong foundation for future research, with the potential to extend its application to other cancer types.
We have expanded the Discussion section (lines 348–353, lines 362–365) to address these limitations and delineate potential areas for future research.
[Revised manuscript]
[lines 348–353]
Despite some markers, such as EPCAM, being broad-spectrum, their unique expression patterns, combined with other TACT markers, formed a CRC-specific diagnostic panel. While the same 10 markers were previously applied in BC research, distinct expression profiles enabled the construction of cancer-specific panels. For CRC, 8 markers were selected for their significant differential expression, optimizing the diagnostic panel’s specificity and sensitivity.
[lines 362–365]
We acknowledge that the data used in this study were collected from specific healthcare institutions, which may limit the generalizability of our findings. Future studies will include participants from multiple regions and diverse populations to improve data representativeness and reduce selection bias.
Reviewer 3 Report
Comments and Suggestions for Authors
The study is severely impaired in its basic CRC cancer biology foundations limiting its scope to 10 genes and to their transcripts (RNA) quantitative expression in patients blood.
A minimal comprehensive analysis of cancer-driving genes is missing as well as a logical mutational genomic approach in the same specimen (minimal number of fully established cancer-driving genes in CRC not sufficient both in number and biased selection based on a limited number of published evidences; along with single point mutations? Gene rearrangements? pathogenic Isoform variants?)
For this reason, the work resembles a pure extension of ancillary pathology markers study to a liquid biopsy quantitative stage skipping critical sequence alterations for which the fine AI analysis using 5 different computational methods results into a forced projection of a biology-evidences poor approach to the CRC molecular and genomic diagnostics using liquid biopsy with insufficient scientific merit.
The same data, when combined with a genomic approach and highly content cancer-driver gene selection panel in the same specimen could have publication-level interest
Author Response
[Comments] The study is severely impaired in its basic CRC cancer biology foundations limiting its scope to 10 genes and to their transcripts (RNA) quantitative expression in patients blood.
A minimal comprehensive analysis of cancer-driving genes is missing as well as a logical mutational genomic approach in the same specimen (minimal number of fully established cancer-driving genes in CRC not sufficient both in number and biased selection based on a limited number of published evidences; along with single point mutations? Gene rearrangements? pathogenic Isoform variants?)
For this reason, the work resembles a pure extension of ancillary pathology markers study to a liquid biopsy quantitative stage skipping critical sequence alterations for which the fine AI analysis using 5 different computational methods results into a forced projection of a biology-evidences poor approach to the CRC molecular and genomic diagnostics using liquid biopsy with insufficient scientific merit.
The same data, when combined with a genomic approach and highly content cancer-driver gene selection panel in the same specimen could have publication-level interest
[Response]
Thank you for your thorough and constructive feedback. We appreciate your concerns regarding the scope of our study and the need for a more comprehensive approach to CRC molecular diagnostics. We would like to address these points as follows:
The primary aim of this study was to evaluate the diagnostic potential of 10 tumor-associated circulating transcript (TACT) markers for CRC detection in a minimally invasive liquid biopsy setting. While it is well-established that various genomic alterations, such as K-RAS and APC mutations, as well as epigenetic modifications like DNA methylation, serve as important biomarkers for CRC and have even led to the development of diagnostic kits, our research focused on RNA-based markers. TACT markers, including mRNA, microRNA, and noncoding RNA, have been demonstrated to be disease-associated biomarkers and offer complementary insights to DNA-based biomarkers. Unlike genomic DNA mutations, which reflect accumulated changes over time, RNA expression reflects the current biological state of the disease, making it particularly useful for real-time monitoring and assessing disease progression.
We fully recognize that a more comprehensive approach incorporating established CRC cancer-driving genes, mutational data (e.g., single point mutations, gene rearrangements), and pathogenic isoform variants would provide deeper insights into CRC biology. These genomic alterations are critical in CRC development and progression, and their detection has been extensively explored in current diagnostic approaches. Future research will aim to integrate transcriptomic and genomic data to establish a more holistic and robust diagnostic framework.
While this study emphasized RNA-based circulating transcripts as preliminary work, future investigations will aim to incorporate transcriptomic and genomic data to establish a more holistic diagnostic framework. Expanding the marker panel to include additional CRC-specific and driver genes identified from large-scale datasets like TCGA, combined with the recruitment of larger and more diverse multi-institutional cohorts, will improve the robustness and applicability of the findings.As suggested, future studies will explore the integration of genomic approaches, including sequencing-based analyses of CRC cancer-driving genes and mutational events, alongside TACT markers. This will allow for a more comprehensive evaluation of molecular changes associated with CRC. Additionally, we plan to expand the marker panel and datasets by recruiting larger, multi-institutional cohorts and including diverse populations, as well as incorporating additional cancer-specific and driver genes identified from large-scale studies, such as TCGA. By expanding this research, we aim to enhance the diagnostic utility of RNA-based markers while exploring their role in real-time disease monitoring.
We appreciate your valuable suggestions, which have helped us refine our study and identify key areas for improvement. These insights will be incorporated into the revised Discussion section (line 378–387) to acknowledge the limitations of the current study and highlight the potential for integrating genomic data in future investigations.
[Revised manuscript]
“This study primarily focused on evaluating the diagnostic potential of 10 tu-mor-associated circulating transcript (TACT) markers for CRC detection using a liquid biopsy approach. To ensure the specificity of the detected transcripts and to avoid genomic DNA contamination, at least one primer or probe within each marker’s primer-probe set was designed to target exon-exon junctions. This design allows for the precise detection of circulating RNA transcripts while minimizing interference from genomic DNA or un-processed RNA. While these markers were selected based on their biological relevance as circulating RNA biomarkers, we acknowledge that a more comprehensive approach incorporating genomic data, including cancer-driving mutations, gene rearrangements, and pathogenic isoform variants, would provide deeper insights into CRC biology. Future research will explore the integration of transcriptomic and genomic data, enabling a more holistic diagnostic framework. Additionally, expanding the marker panel to in-clude established CRC driver genes and utilizing larger, multi-institutional datasets will further enhance the robustness and applicability of the diagnostic model.”
Round 2
Reviewer 3 Report
Comments and Suggestions for Authors
Modifications in the text overcome the key deficiencies in rational and outcomes of the previous version making the submitted manuscript of interest for the general scientific community